# Determining the Optimal Outcome Measures for Studying the Social Determinants of Health

**DOI:** 10.3390/ijerph17093028

**Published:** 2020-04-27

**Authors:** Peter Muennig, Bruce McEwen, Daniel W. Belsky, Kimberly G. Noble, James Riccio, Jennifer Manly

**Affiliations:** 1Department of Health Policy and Management, Columbia University, New York, NY 10032, USA; 2Harold and Margaret Milliken Hatch Laboratory of Neuroendocrinology, The Rockefeller University, New York, NY 10065, USA; mcewen@mail.rockefeller.edu; 3Department of Epidemiology, Columbia University, New York, NY 10032, USA; db3275@cumc.columbia.edu; 4Neuroscience and Education Teachers College, Columbia University, New York, NY 10027, USA; kgn2106@tc.columbia.edu; 5Low-Wage Workers and Communities Division of MDRC, New York, NY 10281, USA; james.riccio@mdrc.org; 6Physicians and Surgeons, Columbia University, New York, NY 10032, USA; jjm71@cumc.columbia.edu

**Keywords:** randomized-controlled trial, social policies and health, anti-poverty policies and health, social determinants of health, outcome measures

## Abstract

Americans have significantly poorer health outcomes and shorter longevity than citizens of other industrialized nations. Poverty is a major driver of these poor health outcomes in the United States. Innovative anti-poverty policies may help reduce economic malaise thereby increasing the health and longevity of the most vulnerable Americans. However, there is no consensus framework for studying the health impacts of anti-poverty social policies. In this paper, we describe a case study in which leading global experts systematically: (1) developed a conceptual model that outlines the potential pathways through which a social policy influences health, (2) fits outcome measures to this conceptual model, and (3) estimates an optimal time frame for collection of the selected outcome measures. This systematic process, called the Delphi method, has the potential to produce estimates more quickly and with less bias than might be achieved through expert panel discussions alone. Our case study is a multi-component randomized-controlled trial (RCT) of a workforce policy called MyGoals for Healthy Aging.

## 1. Highlights

### 1.1. What Do We Already Know about This Topic? 

Poverty is associated with a greater burden of disease in the US than smoking and obesity combined, so anti-poverty policies could have large impacts on population health.

### 1.2. How Does Your Research Contribute to the Field? 

There are a large number of social policy experiments conducted each year in the US, but it is difficult to measure their impacts on health. We provide a framework for doing so.

### 1.3. What Are Your Research’s Implications towards Theory, Practice, or Policy? 

We show that there are a number of survey measures and a handful of biological measures that can reliably be used to measure the health impacts of anti-poverty policies in randomized-controlled trials of social policies.

## 2. Background

Aristotle postulated that the social environment is an important determinant of human health [1]. In the mid-nineteenth century, Rudolf Virchow conducted the first epidemiologic investigation showing the relationship between social conditions and health, documenting what would later come to be known as the “social determinants of health” [2,3]. The 20th century saw social policy and health policy begin to converge in Europe and Latin America [4,5], with the World Health Organization formally recognizing the social determinants of health as “the conditions in which people are born, grow, live, work and age” [3]. The Affordable Care Act in the United States led to billions of dollars spent on the social determinants of health in the hope of preventing disease before it occurs [6,7]. 

One way to address the social determinants of health is to make investments in proven social policies in an attempt to improve the social circumstances of low-income families and individuals [8]. Were it possible to make even a small dent in the burden of disease associated with poverty, enormous gains in health and longevity could be realized [9]. Unfortunately, very few experimental tests of social policies have been conducted [10], and fewer still contain health outcomes [11,12,13]. Only three randomized-controlled trials (RCTs) to date have attempted to collect objective biological measures of health using laboratory or medical examination data [13,14,15,16]. This is in part because there is a good deal of uncertainty surrounding which measures to use [12,17]. 

This uncertainty arises from three challenges. First, the process of biological “wear and tear” associated with living in poverty begins from the time of conception [18,19], but most anti-poverty programs are targeted toward older children and adults [13]. As a result, it is necessary to devise a sensitive measure of health—a “yardstick”—that will change within the time frame of the study. A second challenge is that there is a latency between the time that one is exposed to a social policy and the point at which measurable changes in health occur [20]. While income support programs can immediately lift a family out of poverty, it takes time for that family to invest the money in healthy food, housing, and health care. Even after these investments are made, the effects of these investments take time to manifest as measurable changes in health [14]. A third challenge lies in the nature of the available objective biomarkers of chronic disease [21]. These can be difficult to collect, are sometimes statistically noisy, and may have an inadequate empirical base linking the measure to tangible health outcomes in humans. They also must be conceptually aligned with the expected mechanisms through which a given social policy influences health. What is needed is a sensitive but broad biological measure that can serve as a yardstick of changes in health over a relatively short period of time.

In the absence of such a measure, those who study the effect of social policy on health face an enigma. How should we operationalize what the intervention does vis-à-vis changes in human physiology, and how do we know how far in the future we need to measure these changes given the nature of the intervention? This question is underscored by a recent meta-analysis of 38 experimental social policy RCTs, which found that most lacked adequate statistical power to detect a change in the outcome of interest [13]. Had a sensitive measure been available, the statistical power would have been higher.

The authors set out to build a conceptual model for an ambitious social policy experiment called MyGoals for Healthy Aging. Prior to the exercise, each expert submitted their own conceptual model and relevant outcome measures anonymously. We were surprised to find that each expert in the room had a very different idea about how the intervention might influence health, and how to measure the changes in health that might arise from the intervention. In this paper, we describe our journey toward coming to a consensus about the correct way to measure health outcomes for this health policy experiment.

### 2.1. Measure Selection in Social Policy RCTs

Given that a very long time must pass between changes in one’s social circumstances and distal disease outcomes (like heart disease), measure selection is limited to outcomes than can reasonably change over a short period. These might include psychological stress, mental health, and biological measures of disease risk (“biomarkers”). Biomarkers can include blood assays, such as cholesterol levels, or medical examination data, such as blood pressure. Biomarkers are appealing endpoints for social policy RCTs because they are objective, they have established links to disease, and they change over a relatively short period [22,23]. However, biomarkers must be selected carefully. The challenge is to match the biomarkers to the intervention of interest [12,17].

### 2.2. The Quantitative Shotgun Approach

One approach to rule-in or rule-out a large number of candidate biomarkers is to build a team of experts, often backed up by research staff, to compile and summarize the existing literature. The Stress Measurement Working Group [24,25] and the Targeting Aging with Metformin (TAME) [21] provide two examples of this approach. 

The Stress Measurement Working Group was tasked with selecting measures for several social surveys of aging populations, the Health and Retirement Study in the United States, the English Longitudinal Study of Ageing in England, and the Survey of Health, Ageing and Retirement in Europe [17].

The TAME RCT is designed to investigate whether a drug can slow human aging. Social policy experiments are thought to affect health in a way that is very similar to this drug from a biological standpoint—they alter a cascade of biological events that lead to “wear and tear” on the body’s physiological systems [26]. 

This process (which we might call the “Quantitative Shotgun Approach”) can be very resource intensive, as a good deal of background work must be done to decide how to include or exclude measures, to conduct an extensive literature review, and to manage the information produced by experts over many months of meetings [17,21]. Beyond these logistical challenges, there is also a need to manage bias. The bandwidth of leading experts can be limited, so they might not have the time to read literature that falls outside of their specific area of expertise (which can sometimes be limited to a single biomarker or biological pathway). Open discussions can result in senior, charismatic, or otherwise forceful experts overpowering the ideas of those who are less expressive or who fear professional repercussions of disagreeing with their colleagues [27].

### 2.3. The Delphi Method

An alternative to the Quantitative Shotgun Approach is the Delphi method. The Delphi method also leverages expertise from multiple researchers to identify a consensus on a set of measures. It also uses introductory materials and research as a foundation upon which experts make decisions. However, it has a specific structure designed to overcome some limitations of the Quantitative Shotgun Approach. One advantage is that it is highly structured, thereby reducing tangential discussions [27]. The Delphi method is led by a moderator, and can effectively be conducted over the phone or by email [27]. A much bigger advantage is that key parts of the process are anonymized, so that the ideas of senior researchers are less likely to dominate the discussion.

Deciding which “experts” to select for a given Delphi process really depends on one’s overarching objectives [28]. If the objective is to obtain a collective guess at a parameter in particle physics, then one would draw experts from a very narrow area of physics. However, if one’s objective is to guess at the number of years needed before a given country developed a nuclear bomb, one may wish to draw not just from atomic scientists, but also experts on national security and possibly even anthropologists. Thus, experts might be selected from a very narrow field, more broadly across a given field, or across a broad array of disciplines. 

After an initial briefing that includes an overview of the Delphi method and the problem to be solved, a moderator guides experts through various “rounds” or iterations of the estimate to be obtained (Figure 1). The result is incrementally modified over several sessions. This iterative process allows experts to quickly identify problems in logical thinking or missing information as they constantly reformulate their ideas, ideally arriving at a consensus. 

## 3. The Case Study

Our case study outlines this process for a unique RCT called MyGoals for Healthy Aging, which is a piggyback health study on an innovative welfare intervention called MyGoals for Employment Success [29]. The parent intervention assigned 1798 unemployed recipients of government housing subsidies (Section 8 voucher or public housing) to MyGoals (the treatment group) or to a control group that did not have access to MyGoals. The treatment group is offered three years of employment coaching that uses an explicit methodology for helping participants set and achieve goals across four domains (employment, education/training, financial management, and personal and family well-being) with an explicit focus on identifying and addressing “executive function” challenges that get in the way of goal-achievement in these domains [30]. This coaching is coupled with a package of cash payments that includes a monthly stipend for engaging in substantive coaching sessions plus incentives for achieving certain employment outcomes. Executive function refers to the self-regulation capacities that are essential for successful execution of tasks and include such cognitive skills as stress tolerance, emotional control, time management, metacognition, mental flexibility, task initiation, sustained attention, and others [31,32,33]. If MyGoals increases participants’ earnings and fringe benefits, that increased compensation combined with the financial incentives (which are disregarded in government transfer benefit calculations) and benefits associated with employment, such as the Earned Income Tax Credit and, in our case, funding for the Coronavirus Aid, Relief, and Economic Security (CARES) Act, will improve participants’ net disposable income and reduce their likelihood of being poor or the extent of their poverty relative to the control group. (These benefits arise because the participant is more likely to be employed and therefore filing taxes.).

### 3.1. Methods

We first contacted a number of other investigators of social policy RCTs to ask how they had selected measures in their previous studies. Standard practice, we learned, is to simply query experts, who then provide input as to a range of survey, medical examination, and blood measures to fit the conceptual model of the intervention [34]. The initial conceptual model for MyGoals for Healthy Aging (Figure 2) was simply drawn out by a handful of experts in the social determinants of health. Measures were then selected for each component of the model and tested to statistical power to determine whether they were feasible. This model was included in a grant to the National Institute on Aging, but reviewers disliked the structure and raised questions about the measures we collected as well as the timing of measure collection. Our team was awarded funding to refine the structure, and we decided to deploy the Delphi method to address these reviewer concerns.

### 3.2. Expert Selection

The experts were carefully selected to ensure: (1) comprehensive expertise in the domains of the intervention, (2) the social determinants of health, and (3) familiarity with social policy or experimental research designs. This required not only selecting interdisciplinary experts, but also those who had specifically applied their work across disciplines. The principal investigator and the National Institute on Aging program officer were both familiar with experts who met these characteristics. Six experts were invited, but one could not attend in person, leaving a total of 5. 

The final panel of experts included experts in the social determinants of health within the fields of behavioral psychology (Richard Yi), neuropsychology (Jennifer Manly), neuroendocrinology (Bruce McEwen), aging (Daniel Belsky), and neuroscience (Kimberly Noble). The facilitator was an expert on the social determinants and on health policy (Peter Muennig), and had familiarity with all of the measures discussed. Drs. McEwen, Noble, and Belsky had all conducted research using biological markers relevant for stress and also survey measures of the stress response. Drs. Yi, Manly, and Noble had extensive experience with cognitive measures and familiarity with stress measures. 

In addition to these 5 experts, James Riccio was present. Dr. Riccio, the PI for the parent MyGoals demonstration, has led numerous multi-center social policy RCTs and was therefore invaluable to the de-anonymized discussions as were other MDRC staff. 

### 3.3. The Delphi Process

#### 3.3.1. Stage One

Experts were first distributed a 3-page description of the MyGoals RCT, including its sample size, locations, and details of the intervention. The description was followed by a task (sketching out the conceptual model), two quantitative questions, and another task (linking survey or biological outcome measures to each component of the conceptual model). Prior to the meeting, the facilitator re-drew pen and paper images of each pathway presented by the participants to help preserve the anonymity of the participants and to provide consistency in the presentation.

#### 3.3.2. Stage Two

Participants were brought together to an in-person, 5 h meeting. MDRC staff reviewed the intervention and took questions that came up regarding the experiment. After this introduction, participants discussed the questions from the prior round. A majority opinion was reached among 4 out of the 5 participants during this round, even though the in-person meeting was not meant to develop a consensus.

#### 3.3.3. Stage Three to Five

Participants were contacted following the meeting and asked to anonymously confirm their individual responses. One participant disagreed with the results of the first round. The moderator relayed individual responses and concerns to this participant over 3 additional rounds of email, and the final model was sent to the entire panel for a final vote.

## 4. Results

### 4.1. Round One: Presentation of the Model and Initial Thoughts

Income is thought to be a key pathway. Four out of five participants thought that income was important for health (one participant felt that income was the only important mediator in the pathway between intervention and health outcomes). Three out of five thought that employment was important for health and an equal number thought that the executive function coaching was important for health.

A range of biomarker measures are proposed as potential endpoints. With respect to outcomes, four out of five of the figures the experts submitted had depicted stress reduction as a mechanism through which MyGoals for Healthy Aging would improve physical and mental health outcomes. Three out of five recommended the Perceived Stress Scale and an equal number recommended a biological measure of stress (blood pressure, hair cortisol, or C-reactive protein). Three out of five thought that cognition (measured both using the Behavior Rating Inventory of Executive Function (Adult Version) and tasks from the NIH Toolbox, a set of tests available from the National Institutes of Health) as well as mental health/well-being (measured using the Patient Health Questionnaire 9 and Beck Anxiety Inventory) would be enhanced by the intervention. Two participants thought that body-mass index (BMI) and blood pressure would be decreased and one thought that sleep would be improved. 

The pathways that all 5 of the participants initially sketched out were sequential (e.g., income leads to reduced stress which then leads to increased sleep). 

### 4.2. Round Two: Face-to-Face Meeting and Iterative Critique

For the second round, which was held face-to-face, the facilitator presented tabulated data from the first round of anonymous inputs. After this, each participant’s anonymized conceptual model was reviewed in front of the group. A more in-depth discussion was then undertaken regarding each pathway. This portion of the discussion was for idea generation and was not anonymized. It was meant to exploit the power of free association within the group before returning to anonymized input.

Executive function coaching and health: evaluating potential mechanisms suggests new outcome measures. After the question and answer session, all participants agreed that executive function was important for health in theory, but most felt that the executive function coaching component of the intervention would only be beneficial for some participants. 

This led the participants to probe the MDRC staff regarding the nature of the intervention. Participants were told that the coaches were trained to be supportive rather than prescriptive. That is, rather than saying, “I have the perfect job for you,” they would ask, “are you interested in looking for a job?” They also learned that the coaches were from similar backgrounds as the participants (e.g., had also been recipients of public housing), and were provided an example that one participant had identified the coach as his only “real friend.” They also learned that some participants sought additional coaching sessions for which they were not compensated. These factors led the experts to believe that having a “friendly face” with whom to talk about life’s challenges would potentially have an impact on measures of stress, anxiety, and depression. It was therefore postulated that the coaching had both the effect of addressing some of the executive function deficits of some participants and also serving to provide social support for others.

Measures not identified at the outset of the process were raised as relevant, including sleep quality and perceived discrimination in the workforce. With respect to the latter, all participants came to the conclusion that employment could produce both positive and harmful effects. On the positive side, it affords new opportunities for cognitive engagement, development of social skills (e.g., with co-workers and customers), and to build social capital. On the other hand, participants felt that some aspects of all work, and low-wage work especially, tended to be stressful, and that exposing participants to the employment market would also expose them to discrimination (particularly if they are women or racial/ethnic minorities). Occupational safety was brought up as a potential health hazard specific to being employed.

Timing and model building. A final issue concerned timing of outcome collection. All experts suggested the same general follow-up range: 5−6 years. Therefore, this was not discussed in additional rounds. However, the experts pointed out that it is not possible to actually sequence these events in time, and that sequencing should not be built into the conceptual model (Figure 3).

The group came to the conclusion that the best approach would be to consider the intervention to include executive function, a friendly face, income support, and employment. It was decided that the study should include a measure of stress, a measure of anxiety, a measure of depression, a measure of sleep, and a set of biomarker measures that reasonably serve as intermediate for stress, inflammation, and future disease. The experts argued for established measures that have been traditionally used in social determinants research rather than those that reflect newer research or ideas that might be more controversial for a NIH review panel. For cognitive outcomes, it was felt that both validated survey instruments and data from tasks should be collected.

### 4.3. Rounds Three to Five

The final rounds were conducted by email, with each participant providing their final vote. One participant disagreed with the other four with respect to the physiological measures. This expert expressed the desire to use hair cortisol and telomere length as outcome measures. The moderator and other experts then discussed these ideas by emails. Concerns included: 1) difficulty obtaining hair from bald people or women who are concerned about cosmetic changes [35], and 2) concerns that cortisol levels will be suppressed in some people with chronic stress but elevated in others, leading to results that are difficult to interpret. At that point, Participant 5 agreed that hair collection would be an issue. This reviewer also suggested that a waist-to-hip ratio be obtained as a measure of obesity, noting recent evidence that body-mass index was an incomplete measure of obesity [36,37]. This participant also suggested a long-term measure of life trauma in order to better understand whether trauma altered participants’ responses to the intervention [38]. The final model appears in Figure 3.

## 5. Conclusions

We demonstrate a process by which a conceptual model and outcome measures (along with other useful estimates, like timing for data collection) can be quickly accomplished via the use of expert input. While there is no counterfactual “ideal” against which the results of the Delphi process can be compared, it is worthy to note that the experts produced a significant alteration to the way that the intervention was perceived to work, produced major alterations to the presentation of the conceptual model, and altered four of the original outcome measures. 

Our model has now been incorporated into the research plan for MyGoals for Healthy Aging, and it will be used to determine the selection and timing of the outcome measures if the study is funded by the National Institute on Aging.

We also hope that our study will serve two uses for the greater scientific community. First, we hope that it will provide a foundation for thinking through the ways that social policies might impact health. Second, we hope that it provides a scaffolding and method for the rapid and inexpensive development of future social policy research that includes health outcomes. 

## Figures and Tables

**Figure 1 ijerph-17-03028-f001:**
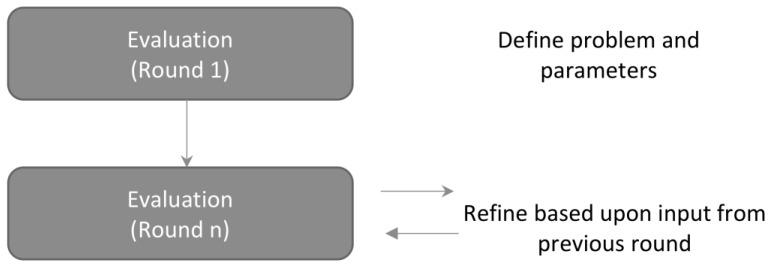
A schematic of the Delphi method. In the Delphi method, experts anonymously discuss a problem over various rounds, with each round, the estimate is refined until consensus or near consensus is reached.

**Figure 2 ijerph-17-03028-f002:**
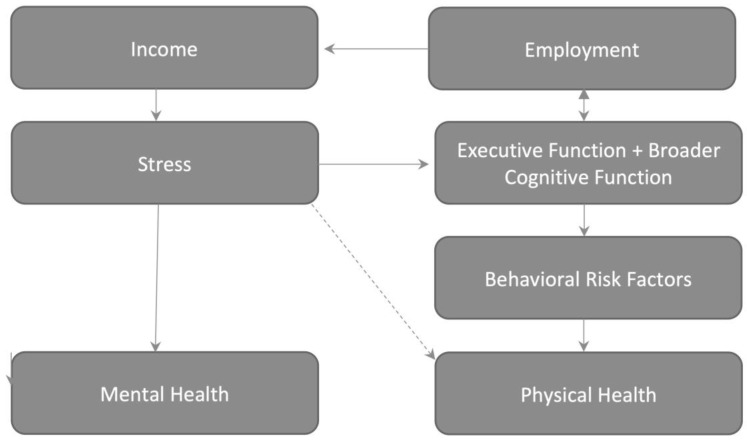
The conceptual model used prior to the Delphi method. When this model underwent review in the National Institute on Aging, reviewers were concerned about the sequencing of events, and asked that it be revised. In this model, income is derived from incentive payments and employment. This increases income thereby reducing psychological stress [13]. Reductions in psychological stress influence physical and mental health via allostatic load [22] while also producing synergies with the executive function training program to reduce neural damage and improve executive function, thereby improving work performance and behavioral risk factors [30,31]. Here, executive function (e.g., the ability to plan and execute those plans) was separated from broader cognitive function to show how employment can enhance broader cognitive skillsets, such as math [31].

**Figure 3 ijerph-17-03028-f003:**
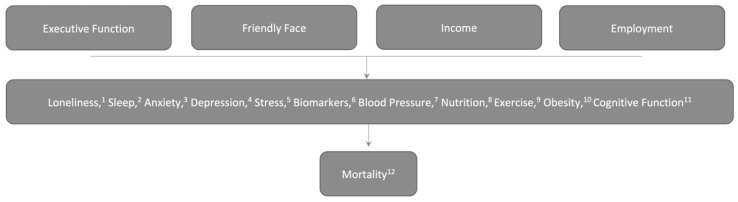
The final model. In this model, the expert panel felt that it was important to greatly simplify the model in response to reviewer concerns about the original model by removing temporal sequencing, including all of the measured outcomes, and to consider enhancements to broader cognitive function as a part of the outcome, rather than part of the intervention (Figure 2). In addition, after learning about the experiences that the executive function coaches have had with the clients, the expert panel felt that part of the intervention entailed adding a “friendly face,” or a friend to talk with about the participants’ problems. (1) Measured using the Three-Item Loneliness Scale; (2) measured using the Insomnia Severity Index; (3) measured using the Beck Anxiety Inventory; (4) measured using the Patient Health Questionnaire 9; (5) measured using the Perceived Stress Scale; (6) Blood Pressure, C-Reactive Protein, Interleukin-6, Hemoglobin A1c; (7) measured by trained examiner three times; (8) measured using the Eating at America’s Table Survey; (9) measured using questions taken from the Behavioral Risk Factor Surveillance System; (10) measured height, weight, waist circumference, hip circumference, waist-to-hip ratio; (11) measured using the Behavior Rating Inventory of Executive Function (BRIEF) and the Flanker + Dimensional Card Sort tasks from NIH Toolbox; (12) measured using the National Death Index. **Note:** Serum will be banked for possible future biomarker analyses such as conserved transcriptional response to adversity, gene X environment studies, biological clock studies, and metabolomic studies as these are rapidly evolving fields of study that will undoubtedly change over the period of performance of the grant.

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
