# Peer review of "Determining the Optimal Outcome Measures for Studying the Social Determinants of Health"

_ijerph, 2020, doi:10.3390/ijerph17093028_

Round 1
Author Response
Reviewer 1
Comments on IJERPH770426 Determining the optimal outcome measures for mericans2 studying the social determinants of health
Summary: This study used a systematic process, called the Delphi method, to determine the optimal measures of health in experimental social policy randomized-controlled trials (RCTs) to check whether the intervention was perceived to work in improving health and wellbeing. This paper describes a case study with focuses on a multi-component randomized-controlled trial (RCT) of a workforce policy called MyGoals for Healthy Aging.
It is well-known in the early literature that poverty is associated with a greater burden of disease than smoking and obesity combined and anti-poverty policies could have large impacts on population health. Recently, in the US, the passage of the Affordable Care Act has meant that billions in dollars that might have gone to medical care to treat illness and disability are now being spent on preventing disease via broad social policies instead. This paper contributes to this new development.
The strength of this paper: This is a well written methodology discussion paper. The Delphi method was thought to have the potential to produce estimates more quickly, with less bias, and with more accuracy than might be achieved through expert panel discussions alone, which could be very time consuming, hence restricted by the inflexibility of leading experts.
The Delphi method leverages expertise from multiple researchers to identify a consensus set of measures over the phone or by email based on their previous experiences in social determinants of health . A much bigger advantage is that key parts of the process are anonymized, so that the ideas of senior researchers are less likely dominate the discussion. In addition, the experts were carefully selected to ensure: 1) comprehensive expertise in the domains of the 129 intervention, and 2) familiarity with social policy or experimental research designs.
Three stages are used to collect the responses and thoughts: firstly introducing the project and collecting initial thoughts by phone or emails, secondly having an in-person 5 hour in-depth discussion to response questions, and lastly confirming their individual responses by phone or emails. Inquiries and comments:
- (1) Figure 2 did not present major findings from the existing literate on social determinants of health, such as no indicative relationships shown between employment and stress, income and behavioral risk factors; behavioral risk factors and mental/physical health, etc.
Response. Thank you. We have added citations to Figure 2. We have also corrected an error in the figure in which an arrow was missing between behavioral risk factors and physical health. Note that Figure 2 was the foundational model with which the NIH reviewers found fault. Therefore, it is included only to give the reader a sense of how the expert input changed the presentation of the model.
- (2) What is the meaning of “executive function”? It has been placed together with Broader cognitive function in Figure 2. But in Figure 3, it has been placed separately as a social policy while cognitive function as health outcome.
Response. Thank you for catching this. We have made corrections in the figure and in the text. (See lines 141-145.) The expert panel had advised moving broader cognitive function from the top of the diagram to the bottom so that reviewers would not confuse components of the intervention with those of the outcomes.
- (3) What is the meaning of “friendly face”? only having someone to talk friendly on life challenges?
Response. Yes. We have now added this to the figure and additional information to the text.
- (4) In figure 3, why education, equality, social trust, safety and perceived discrimination are not included as important social policies?
Response. Thank you we have modified the text below the figure. The MyGoals for Healthy Aging RCT does not intervene on education. By encouraging workforce participation, it can influence perceived discrimination. This was recommended by the expert panel as a variable to test for mediation and interaction terms. (Please see Figure 3, bottom of page.)
- (5) In figure 3, sleep, nutrition and exercise are all healthy and/or risk behaviours, which have impacts on health outcomes, but they are not healthy outcomes. Should these be listed separately in Figure 3 as healthy behaviours?
Response. The challenge that we face in the MyGoals for Healthy Aging RCT is that we are unable to sequence the measures. Rather, all measures will be collected 5 years following randomization. Therefore, we have to measure sleep, nutrition, etc., as distal outcome measures rather than as mediators of health outcomes. We have better described this issue in the text. (Please see lines 246-257.)
- (6) In figure 3, why stress and inflammation are together as outcomes?
Response. This is how the expert panel conceptualized these measures. However, we agree that this is confusing and have separated them. (Please see Figure 3, bottom of the figure.)
- A systematic literature reviews on WHO definition of healthy ageing , and on social determinants of health are necessary and will help more on foundation thinking and the determinants and relationships presented and selected.
Response. Thank you. These have been included in the background section. (Please see lines 40-81.)
Reviewer 2 Report
It's very important to measure social determinants of health(SDH). Unfortunately, (1) I wonder whether this study has a useful work about the SDH's literatures. (2) What's more, I wonder the relationship bewteen the RCT and Delphi. (3) In this study, the results could be guilty of biased reporting. Because all investigators are experts in the social determinants of health within the fields of behavioral psychology , neuropsychology, neuroendocrinology, aging, neuroscience. The investigators with disease or lack of health knowledge would be ignored. (4) Besides, the background in this article is so short and bad that I cann't understand why and what is the optimal outcome measures for studying the social determinants of health.
Author Response
Reviewer 2.
It's very important to measure social determinants of health(SDH). Unfortunately,
- I wonder whether this study has a useful work about the SDH's literatures.
Response. We have now added information to the background section that includes the WHO definition of SDH and relevant citations. (Please see lines 40-77.)
- What's more, I wonder the relationship bewteen the RCT and Delphi.
Response. We have added information in the background section to help clarify this point. (Please see lines 78-84). Our RCT requires a conceptual model to guide the collection of appropriate outcome measures. The Delphi process was used to obtain information on which outcome measures are most appropriate to include in the RCT given the nature of the intervention.
- In this study, the results could be guilty of biased reporting. Because all investigators are experts in the social determinants of health within the fields of behavioral psychology , neuropsychology, neuroendocrinology, aging, neuroscience. The investigators with disease or lack of health knowledge would be ignored.
Response. This bias is intentional. The Delphi method was not conducted to obtain outcomes from the underlying RCT. Rather, it was conducted to guide measure selection for the underlying RCT. The universe of appropriate measures is limited, and requires deep expertise across a known number of domains. We selected one expert from each of these domains to help design the model. (See lines 125-131.)
- Besides, the background in this article is so short and bad that I cann't understand why and what is the optimal outcome measures for studying the social determinants of health.
Response. Thank you, we have re-written the background section of the paper.
Reviewer 3 Report
Please see attached document.

Author Response
Reviewer 3.
This is a beautifully written, important, and interesting article. It provides a compelling case example of the effective use of the Delphi method to inductively elicit appropriate outcome measures, data collection timelines, and an underlying theory of change for a social policy-based intervention designed to improve health. I have minor suggestions to maximize the already considerable contributions of this manuscript.
- Since one of the themes of the paper is the value of the Delphi method, it would be helpful to have an additional sentence or two that grounds this specific application of that method in the larger literature about it. In particular, I would like to see a bit more detail about the ways in which this expert selection process conforms to standard practice.
Response. Thank you. We have re-written parts of the background and have added substantially more information on the Delphi process, including the selection of experts in this case. (See lines 125-131.)
- The fact that this study produced a refined understanding of the ways in which the intervention works is a powerful example of the need to have a well-articulated program theory. The authors’ particular finding suggests some implications for the implementation and fidelity protocols for this new program. Will there be next steps to make sure that such alterations are built into the roll-out of the program?
Response. Thanks for this observation. We have now included a “next steps” section in the discussion. (See line 277.)
- That’s it, other than a typo in line 148 (“liking” should be “linking”). Great article. I look forward to reading and using this article when it appears in print.
Response. Many thanks! We have corrected this typo and have made all of the suggested revisions.
Round 2
Reviewer 1 Report
I am satisfied with the revision, I have no further comments.
Reviewer 2 Report
The article would be accepted because the authors have made every effort.